# UNICORN: CONTINUAL LEARNING WITH A UNIVERSAL, OFF-POLICY AGENT

## ABSTRACT

Some real-world domains are best characterized as a single task, but for others this perspective is limiting. Instead, some tasks continually grow in complexity, in tandem with the agent's competence. In *continual learning* there are no explicit task boundaries or curricula. As learning agents have become more powerful, continual learning remains one of the frontiers that has resisted quick progress. To test continual learning capabilities we consider a challenging 3D domain with an implicit sequence of tasks and sparse rewards. We propose a novel agent architecture called *Unicorn*, which demonstrates strong continual learning and outperforms several baseline agents on the proposed domain. The agent achieves this by jointly representing and efficiently learning multiple policies for multiple goals, using a parallel off-policy learning setup.

## 1 INTRODUCTION

*Continual learning*, that is, learning from experience about a continuing stream of tasks in a way that exploits previously acquired knowledge or skills, has been a longstanding challenge to the field of AI (Ring, 1994; Schaul, 2018). A major draw of the setup is its potential for a fully autonomous agent to incrementally build its competence and solve the challenges that a rich and complex environment presents to it – without the intervention of a human that provides datasets, task boundaries, or reward shaping. Instead, the agent considers tasks that continually grow in complexity, as the agent's competence increases. An ideal continual learning agent should be able to (A) solve *multiple* tasks, (B) exhibit synergies when tasks are *related*, and (C) cope with deep *dependency* structures among tasks (e.g., a lock can only be unlocked after its key has been picked up).

A plethora of continual learning techniques exist in the supervised learning setting, as recently reviewed by Parisi et al. (2018), who note that more research is needed to tackle continual learning with autonomous agents in uncertain environments - a setting well-suited to Reinforcement Learning (RL). Previous work on continual learning with RL, and specifically on solving tasks with deep dependency structures, has typically focused on *separating learning into two stages*; that is, first individual skills are acquired separately and then later recombined to help solve a more challenging task (Tessler et al., 2017; Oh et al., 2017). Finn et al. (2017) framed this as meta-learning and trained an agent on a distribution of tasks that is explicitly designed to adapt to a new task in that distribution with very little additional learning. In a linear RL setting Ammar et al. (2015) learn a latent basis via policy gradients to solve new tasks as they are encountered. Brunskill & Li (2014) derive sample complexity bounds for option discovery in a lifelong learning setting.

In this work we aim to solve tasks with deep dependency structures using *single-stage end-to-end learning*. In addition, we aim to train the agent on *all* tasks simultaneously regardless of their complexity. We argue that, in order to tackle this problem, we need two basic ingredients: *task generalization* and *off-policy learning*. This allows experience to be shared across tasks, enabling the agent to efficiently develop representations and competency for each task in parallel.

There are a number of RL techniques that perform task generalization by incorporating tasks directly into the definition of the value function (Sutton et al., 2011; Dieterich, 2000; Kaelbling, 1993). Universal Value Function Approximators (UVFAs; Schaul et al. (2015)) are the latest iteration of these works that effectively combines the value functions for multiple rewards and policies into a single function approximator. We extend UVFAs with off-policy learning (Sutton et al., 2011; Ashar,

1994; Peng & Williams, 1994) about multiple goals simultaneously, and scale them up to a parallel agent architecture (Mnih et al., 2016; Espeholt et al., 2018) and train them end-to-end (they were originally trained in a two-stage process). The resulting continual learning agent, called *Unicorn*[1], is capable of consistently solving continual learning tasks with deep dependency structures, at scale, in complex domains (Figure 1). It does this by sharing experience and reusing representations and skills across tasks. We claim the following contributions:

- Unicorn: The first effective end-to-end training of UVFA at scale, off-policy, using a novel combination of state-of-the-art RL architectures.
- Adapting the parallelized policy-based state-of-the-art RL architecture (Espeholt et al., 2018) to value-based as well as incorporating off-policy learning and off-policy corrections into the architecture.
- A detailed investigation with multiple ablation experiments, showing that Unicorn effectively learns multiple tasks in parallel and exhibits synergies when tasks are related.
- A continual learning experiment whereby the Unicorn learns to solve tasks with deep dependency chains (e.g., collect a key, unlock a lock, open a door, and collect a chest in that order. Then receive a reward upon completion of the task). This is achieved using task generalization and off-policy learning, to solve this task which was impossible for existing methods (see the 'expert(chest)' baseline in Figure 1, bottom).
- Treasure world, a new continual learning domain with deep dependency structure that retains the full complexity of Beattie et al. (2016)'s DM Lab (action space, vision, memory, navigation). This is especially important since continual learning benchmarks for RL are scarce (Schaul, 2018).

## 2 BACKGROUND

**Reinforcement learning** (RL) is a computational framework for making decisions under uncertainty in sequential-time decision making problems (Sutton & Barto, 1998). An RL problem is formulated as a Markov decision process, defined as a 5-tuple $\langle \mathcal{S}, \mathcal{A}, r, \mathcal{P}, \gamma \rangle$ where $\mathcal{S}$ is a set of states, $\mathcal{A}$ is a set of actions, $r : \mathcal{S} \times \mathcal{A} \to \mathbb{R}$ is the reward function, $\mathcal{P} : \mathcal{S} \times \mathcal{A} \times \mathcal{S} \to [0, 1]$ is a transition probability distribution and $\gamma \in [0, 1)$ is a discount factor. A policy $\pi$ maps states $s \in \mathcal{S}$ to a probability distribution over actions. We define the *return* from a given time step $t$ as the discounted sum of rewards: $R_t = r_{t+1} + \gamma r_{t+2} + \gamma^2 r_{t+3} + \cdots = \sum_{k=0}^{\infty} \gamma^k r_{t+k+1}$, where $r_t = r(s_t, a_t)$. Action value functions $Q^\pi(s, a) = \mathbb{E}^\pi[R_t | s_t = s, a_t = a]$ estimate the expected return for an agent that selects an action $a \in A$ in some state $s \in S$, and follows policy $\pi$ thereafter. The optimal action value function $Q^*(s, a)$ estimates the expected return with respect to the optimal policy $\pi^*$.

**Q-learning** (Watkins, 1989; Watkins & Dayan, 1992) can be used to estimate the optimal action value function $Q^*(s, a)$, via an iterative bootstrapping procedure in which $Q(s_t, a_t)$ is updated towards a *bootstrap target* $Z_t$ that is constructed using the estimated Q-value at the next state: $Z_t = r_{t+1} + \gamma \max_a Q(s_{t+1}, a)$. The difference $\delta_t = Z_t - Q(s_t, a_t)$ is referred to as the temporal difference (TD) error (Sutton & Barto, 1998).

**Multi-step Q-learning** variants (Watkins, 1989; Peng & Williams, 1994; De Asis et al., 2017) use multiple transitions in a single bootstrap target. A common choice is the n-step return defined as $G_t^{(n)} = \sum_{k=1}^{n} \gamma^{k-1} r_{t+k} + \gamma^n \max_a Q(s_{t+n}, a)$. In calculating n-step returns there may be a mismatch in the action selection between the target and behavior policy within the $n$ steps. In order to learn off-policy, this mismatch can be corrected using a variety of techniques (Sutton et al., 2009; Maei et al., 2010; Precup et al., 2001; Munos et al., 2016). We deal with off-policy corrections by truncating the returns whenever a non-greedy action is selected, as suggested by Watkins (1989).

**Universal Value Function Approximators** (UVFA) extend value functions to be conditional on a goal signal $g \in \mathcal{G}$, with their function approximator (such as a deep neural network) sharing an internal, goal-independent representation of the state $f(s)$ (Schaul et al., 2015). As a result, a UVFA $Q(s, a; g)$ can compactly represent multiple policies by conditioning on any goal signal $g$ and choosing actions greedily. UVFA's have previously been implemented in a two-stage process

---

[1]*Unicorn* stands for "UNIversal Continual Off-policy Reinforcement learNing".

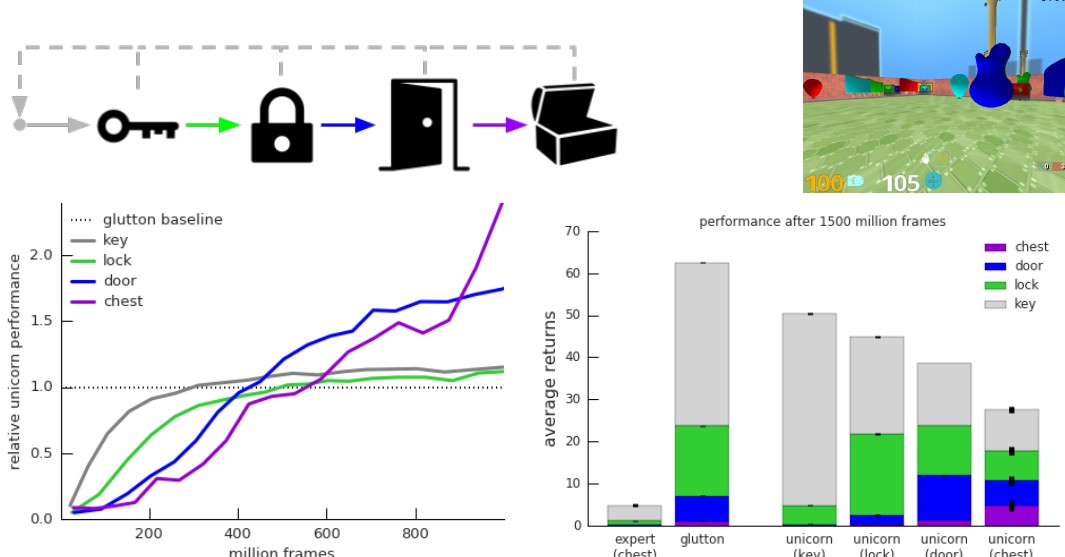

Figure 1: **Top: Continual learning domain**. the agent needs to find and pick up objects in the specific order `key`, `lock`, `door`, `chest` (diagram, top left), while acting in the rich 3D environment of DM Lab (screenshot, top right). Any deviation from that order resets the sequence (dashed gray lines). **Bottom: Results**. The objective is to pick up as many chests as possible (purple). The bottom left plot shows performance relative to final performance of the best baseline (glutton, dashed line) as a function of training. Our method quickly learns to become competent on all 4 subtasks, one at a time. The margin of improvement over the baseline is largest on the hardest subtasks (see section 4.5 for details). The stacked bar plot (bottom right) shows the average number of collected objects after training (1.5B frames). The unicorn can be conditioned on 4 goals, each conditioning shown as a separate stack. E.g. the Unicorn's `door` policy typically collects 15 `key`s, 12 `lock`s, 10 `door`s and 1 `chest`. Note that the strict ordering imposed by the domain requires that there are always at least as many `key`s as `lock`s, etc. Black bars indicate the standard deviation across 5 independent runs.

involving a matrix factorization step to learn embeddings and a separate multi-variate regression procedure. In contrast, the Unicorn learns $Q(s, a; g)$ end-to-end, in a joint parallel training setup with off-policy goal learning and corrections.

**Tasks vs. goals**. For the purposes of this paper, we assign distinct meanings to the terms **task** ($\tau$) and **goal signal** ($g$). A goal signal modulates the behavior of an agent (e.g., as input to the UVFA). In contrast, a task defines a pseudo-reward $r_\tau$ (e.g., $r_{key} = 1$ if a `key` was collected and 0 otherwise). During learning, a vector containing all pseudo-rewards is visible to the agent on each transition, even if it is pursuing one specific goal. Each experiment defines a discrete set of $K$ tasks $\{\tau_1, \tau_2, \ldots, \tau_K\}$. In transfer experiments, tasks are split between $K'$ training tasks and $K - K'$ hold-out tasks.

# 3 UNICORN

This section introduces the *Unicorn* agent architecture with the following properties to facilitate continual learning. *(A):* The agent should have the ability to simultaneously learn about multiple tasks, enabling domains where new tasks are continuously encountered. We use a joint parallel training setup with a single learner but many actors working on different tasks to accomplish this (sections 3.2 and section 3.4). *(B):* As the agent accumulates more knowledge, we want it to generalize by reusing some of its knowledge to solve related tasks. This is accomplished by using a single UVFA to capture knowledge about all tasks, with a separation of goal-dependent and goal-independent representations to facilitate transfer (section 3.1). *(C):* The agent should be effective in domains where tasks have a deep dependency structure. This is the most challenging aspect, but is enabled by off-policy learning from the experience across all tasks (section 3.3). For example, an actor pursuing the `door` goal will sometimes, upon opening a door, subsequently stumble upon a `chest`. To the door-pursuing actor this is an irrelevant (non-rewarding) event, but when learning about the `chest` task this same event is highly interesting as it is one of the rare non-zero reward transitions.

We will next describe the various components that comprise the Unicorn. The full Unicorn algorithm can be found in Appendix A.

## 3.1 VALUE FUNCTION ARCHITECTURE

A key component of the Unicorn agent is a UVFA, which is an approximator, such as a neural network, that learns to approximate $Q(s, a; g)$. The power of this approximator lies in its ability to be conditioned on a goal signal $g$. This enables the UVFA to learn about multiple tasks simultaneously where the tasks themselves may vary in their level of difficulty (e.g., tasks with deep dependencies). Our proposed UVFA architecture is depicted schematically in Figure 2: the current frame of visual input is processed by a convolutional network (CNN; LeCun et al. (1998)), followed by a recurrent layer of long short-term memory (LSTM; Hochreiter & Schmidhuber (1997)). As in (Espeholt et al., 2018) the previous action and reward are part of the observation. The output of the LSTM is concatenated with an "inventory stack", to be described in section 4.1, to form a goal-independent representation of state $f(s)$. This vector is then concatenated with a goal signal $g$ and further processed via two layers of a multi-layer perceptron (MLP) with ReLU non-linearities to produce the output vector of Q-values (one for each possible action $a \in \mathcal{A}$). The union of trainable parameters from all these components is denoted by $\theta$. Further details about the networks and hyperparameters can be found in Appendix B.

## 3.2 BEHAVIOUR POLICY

At the beginning of each episode, a goal signal $g_i$ is sampled uniformly, and is held constant for the entire episode. The policy executed is $\epsilon$-greedy after conditioning the UVFA on the current goal signal $g_i$: with probability $\epsilon$ the action taken $a_t$ is chosen uniformly from $\mathcal{A}$, otherwise $a_t = \arg\max_a Q(s_t, a; g_i)$.

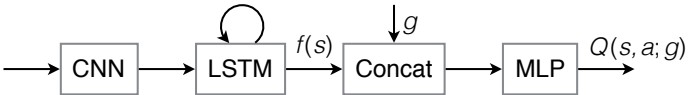

Figure 2: Our UVFA architecture. Observations are processed via a CNN and an LSTM block to produce a goal-independent representation $f(s)$, which is concatenated with the goal signal $g$, and then processed by an MLP into $Q(s, a; g)$.

### 3.3 OFF-POLICY MULTI-TASK LEARNING

Another key component of the Unicorn agent is its ability to learn about multiple tasks off-policy. Therefore, even though it may be acting on-policy with respect to a particular task, it can still learn about other tasks from this shared experience in parallel. Concretely, when learning from a sequence of transitions, Q-values are estimated for all goal signals $g_i$ in the training set and $n$-step returns $G_{t,i}^{(n)}$ are computed for each corresponding task $\tau_i$ as $G_{t,i}^{(n)} = \sum_{k=1}^{n} \gamma^{k-1} r_{\tau_i}(s_{t+k}, a_{t+k}) + \gamma^n \max_a Q(s_{t+n}, a; g_i)$. When a trajectory is generated by a policy conditioned on one goal signal $g_i$ (the on-policy goal with respect to this trajectory), but used to learn about the policy of another goal signal $g_j$ (the off-policy goal with respect to this trajectory), then there are often action mismatches, so the off-policy multi-step bootstrapped targets become increasingly inaccurate. Following (Watkins, 1989), we therefore *truncate* the $n$-step return by bootstrapping at all times $t$ when the taken action does not match what a policy conditioned on $g_j$ would have taken,[2] i.e. whenever $a_t \neq \arg\max_a Q(s_t, a; g_j)$. The network is updated with gradient descent on the sum of TD errors across tasks and unrolled trajectory of length $H$ (and possibly a mini-batch dimension $B$), yielding the squared loss (Equation 1) where errors are not propagated into the targets $G_{t,i}^{(n)}$.

$$\mathcal{L} = \frac{1}{2} \sum_{i=1}^{K'} \sum_{t=0}^{H} \left( G_{t,i}^{(n)} - Q(s_t, a_t; g_i) \right)^2 \ , \tag{1}$$

### 3.4 PARALLEL AGENT IMPLEMENTATION

To effectively train such a system at scale, we employ a parallel agent setup consisting of multiple *actors*[3], running on separate (CPU) machines, that generate sequences of interactions with the environment, and a single *learner* (GPU machine) that pulls this experience from a queue, processes it in mini-batches, and updates the value network (see Figure 3, right). This is similar to the recently proposed Importance Weighted Actor-Learner Architecture agent (Espeholt et al., 2018).

Each actor continuously executes the most recent policy for some goal signal $g_i$. Together they generate the experience that is sent to the learner in the form of trajectories of length $H$, which are stored in a global queue. Before each new trajectory, an actor requests the most recent UVFA parameters $\theta$ from the learner. Note that all $M$ actors run in parallel and at any given time will generally follow different goals.

The learner batches up trajectories of experience pulled from the global queue (to exploit GPU parallelism), passes them through the network, computes the loss in equation 1, updates the parameters $\theta$, and provides the most recent parameters $\theta$ to actors upon request. Batching happens both across all $K'$ training tasks, and across $B \times H$ time-steps ($B$ trajectories of length $H$). Unlike DQN (Mnih et al., 2015), we do not use a target network, nor experience replay: the large amount of diverse experience passing through seems to suffice for stability.

## 4 EXPERIMENTS

Following our stated motivation for building continual learning agents, we set up a number of experiments that test the Unicorn's capability to solve multiple, related and dependent tasks. Specifically, we now briefly discuss how the list of desirables properties presented in section 3 are addressed in

---

[2] Returns are also truncated for the on-policy goal when epsilon (i.e., non-greedy) actions are chosen.

[3] For our transfer experiments, a separate set of actor machines perform evaluation, by executing policies conditioned on the hold-out goal signals, but without sending any experience back to the learner. Further details about the Unicorn agent setup, the hardware used, as well as all hyperparameters can be found in Appendix B.

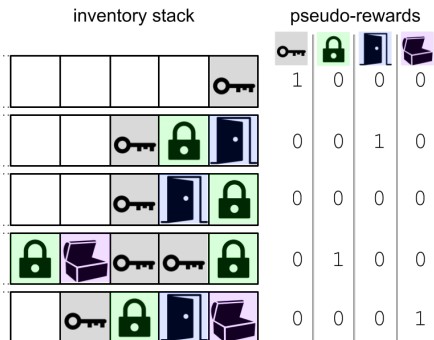 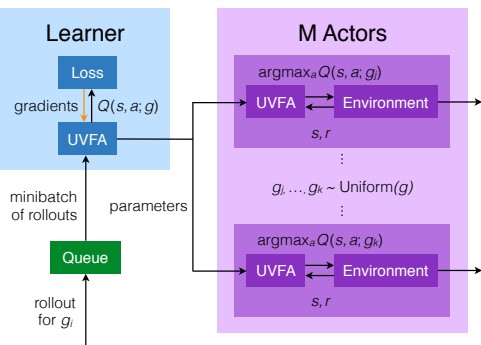

Figure 3: **Left: Inventory stack and pseudo-rewards**. Each row illustrates an event where the agent touches an object. The inventory stack (left) tracks the sequence of recently collected objects, the pseudo-reward vector (right) is used to train the agent, and is 4-dimensional in this case because there are visible 4 tasks. See section 4.5, where pseudo-rewards require the stack to contain the dependent objects in the correct order (`key`, `lock`, `door`, `chest`) – any extra object resets the sequence (like the `door` in row 3). **Right: Agent architecture**. Rollouts of experience are stored in a global queue. Gradients w.r.t. the loss are propagated through the UVFA. After every training step, all $M$ actors are synchronized with the latest global UVFA parameters.

the sections to follow. Section 4.3 measures Unicorn's capability of learning about multiple tasks simultaneously (A), using a single experience stream and a single policy network; each task consists of collecting a different kind of object. The setup in section 4.4 investigates the extent to which knowledge about such tasks can be transferred to related tasks (B), which involve collecting objects of unseen shape-color combinations, for example. Finally, section 4.5 investigates the full problem of continual learning with a deep dependency structure between tasks (C) such that each is a strict prerequisite for the next.

## 4.1 DOMAIN

All of these experiments take place in the *Treasure World* domain. We chose a visually rich 3D navigation domain within the DM Lab framework (Beattie et al., 2016). The specific level used consists of one large room filled with 64 objects of multiple types and equal frequency. Whenever an object is collected, it respawns at a random location in the room. Episodes last for 60 in-game seconds, which corresponds to 450 time-steps. The continual learning experiments last for 120 in-game seconds. The objects used in the multi-task and transfer domains are different color variations of cassettes, chairs, balloons and guitars. For continual learning, the TV, ball, balloon and cake objects play the *functional* roles of a `key`, `lock`, `door` and `chest` respectively. Visual observations are provided only via a first-person perspective, and are augmented with an inventory stack with the five most recently collected objects (Figure 3). Picking up is done by simply walking into the object. There is no special pick-up action; however, pick-ups can be conditional, e.g., a `lock` can only be picked up if the `key` was picked up previously (see Figure 3). The goal signals are pre-defined one-hot vectors unless otherwise stated.

## 4.2 BASELINES

We compare four baselines to the Unicorn agent. Note that we report the average Unicorn performance across all tasks.

We first define the single-task expert that uses the same architecture and training setup as the Unicorn, but acts always on-policy for its single task, and learns only about its single task; in other words, the agent uses a constant goal signal $g$. We train one of these agents for each individual task. From the single task expert, we create two baselines:

The first baseline is *expert (single)*, which is the single-task expert performance averaged across all tasks. The horizontal axis for this baseline is not directly comparable as the experts together

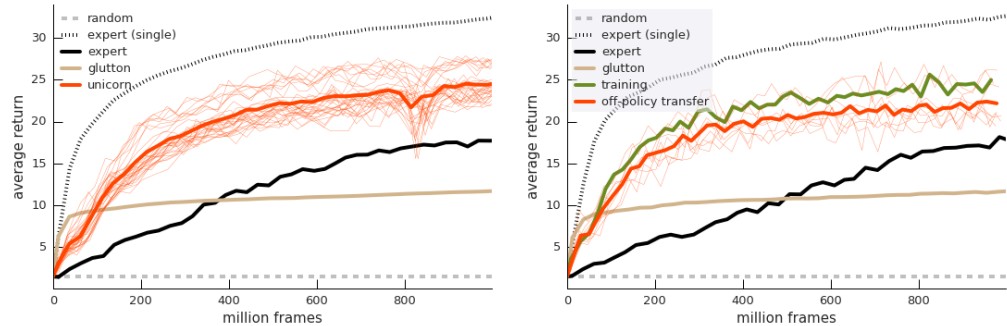

Figure 4: **Left: Multi-task learning**. A single Unicorn agent learns to collect any out of 16 object types in the environment. Each thin red line corresponds to the on-policy performance for one such task; the thick line is their average. We observe that performance across all tasks increases together, and much faster, than when learning separately about each task (black). See text for the baseline descriptions. **Right: Off-policy transfer (I)**: The average performance across the 12 training tasks is shown in green. Without additional training, the same agent is evaluated on the 4 cyan tasks (thin red learning curves): it has learned to generalize well from fully off-policy experience.

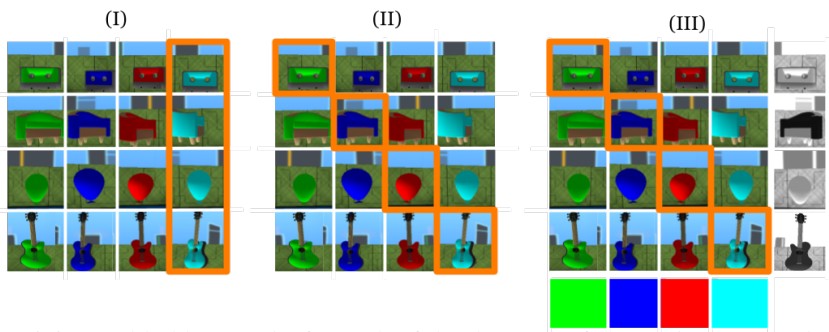

Figure 5: Training and hold-out tasks for each of the three transfer experiments. Each square represents one task that requires collecting objects of a specific shape and color. Hold-out tasks are surrounded by orange boxes. The augmented training set (III) includes the abstract tasks that reward just color or just shape.

consume $K$ times more experience than the Unicorn (as each single-task expert is trained on a separate network). We therefore represent this baseline with a dotted line to indicate an upper performance bound.

The second baseline is *expert*, which focuses on sample complexity and takes all accumulated experience of all the single-task expert agents, across all tasks, into account. In this case, the axes are directly comparable.

The third type of baseline, denoted *glutton*, also uses the same architecture and training setup, but uses a single composite task whose pseudo-reward is the sum of rewards of all the other tasks $r_{glutton}(s, a) = \sum_i^K r_i(s, a)$. This is also a single-task agent that always acts on-policy according to this cumulative goal. Its performance is measured by calculating the rewards the glutton policy obtains on the individual tasks. This baseline is directly comparable in terms of compute and sample complexity, but of course it optimizes for a different objective, so its maximal performance is inherently limited because it cannot specialize. It is nevertheless useful in scenarios where the expert baseline fails to get off the ground (section 4.5). As a fourth baseline, we indicate the performance of a uniformly *random* policy.

## 4.3 LEARNING MULTIPLE TASKS

The multi-task Treasure World experiment uses 16 unique objects types (all objects have one of 4 colors and one of 4 shapes), with an associated task $\tau_i$ for each of them: picking up that one type of object is rewarding, and all others can be ignored. Figure 4 shows the learning curves for Unicorn and how they relate to the baselines; data is from two independent runs. We see that learning works

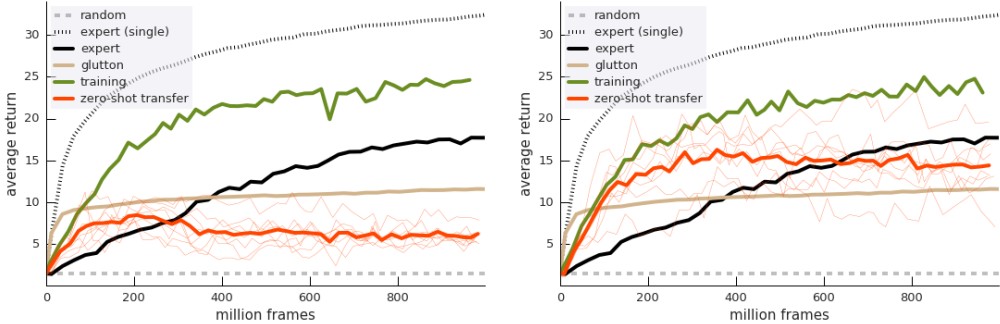

Figure 6: Combinatorial zero-shot transfer. **Left:** during training (green), the Unicorn agent sees only a subset of 12 tasks, and experience from those 12 behavior policies (II). The transfer evaluation is done by executing the policies for the 4 hold-out tasks (without additional training). After an initial fast uptick, transfer performance drops again, presumably because the agent specializes to only the 12 objects that matter in training. Note that it is far above random because even when conditioned on hold-out goal signals, the agent moves around efficiently and picks up transfer objects along the way. **Right:** zero-shot transfer performance is significantly improved when the training tasks include the 8 abstract tasks (III), for a total of $K' = 20$.

on all of the tasks (small gap between the best and the worst), and that learning about all of them simultaneously is more sample-efficient than training separate experts, which indicates that there is beneficial generalization between tasks. As expected, the final Unicorn performance is much higher than that of the glutton baseline.

**Ablative study - scalability:** We conducted an experiment to understand the scalability of the agent to an increasing number of tasks. Figure 8 in the appendix shows results for the setup of $d = 7$ object types: the results are qualitatively the same as $d = 16$, but training is at about twice as fast (for both Unicorn and the baselines), indicating a roughly linear scalability for multi-task learning. Scaling to even larger numbers of tasks is a matter for future work, and some ideas are presented in Section 5.

## 4.4    GENERALIZATION TO RELATED TASKS

In the same setup of 16 object types as above, we conduct multiple transfer learning experiments that test generalization to new tasks. Figure 5 summarizes what the training and hold-out sets are for each experiment.

The first transfer experiment (Figure 5-II) investigates zero-shot transfer to a set of four hold-out tasks (objects within the orange boxes) that see neither on-policy experience nor learning updates. Generalization happens only through the relation in how goal signals are represented: each $g_i \in \mathbb{R}^8$ is a two-hot binary vector with one bit per color and one bit per shape. Successful zero-shot transfer would require the UVFA to factor the set of tasks into shape and color, and interpret the hold-out goal signals correctly. Figure 6 (left) shows the average performance of the hold-out tasks, referred to as *zero-shot*, compared to the training tasks and the additional baselines. We observe that there is some amount of zero-shot transfer, because the zero-shot policy is clearly better than random.

The second transfer experiment (Figure 5-III) augments the set of training tasks by 8 abstract tasks (20 training tasks in total) where reward is given for picking up any object of one color (independently of shape), or any object of one shape (independently of color). Their goal signals are represented as one-hot vectors $g_i \in \mathbb{R}^8$. Figure 6 (right) shows that this augmented training set substantially helps the zero-shot performance, above what the glutton baseline can do. These results are consistent with those of Hermann et al. (2017). More detailed learning curves can be found in the appendix.

**Ablative study - Learning only from off-policy updates:** In a probing experiment (Figure 5-I), the Unicorn actors only act on-policy with respect to the 12 training goal signals (all non-cyan objects), but learning happens for the full set of 16 objects; in other words the cyan objects (surrounded by the orange bounding box) form a partial hold-out set, and are learned purely from off-policy experience. Figure 4 summarizes the results, which show that the agent can indeed learn about

goals from purely off-policy experience. Learning is not much slower than the on-policy goals. Contrasting this with Figure 6 shows that the Unicorn's performance is explained to a large extent by the off-policy learning about many goals, and not by the more diverse generated experience.

## 4.5  CONTINUAL LEARNING WITH DEEP DEPENDENCIES

This section presents experiments that test the Unicorn agent's ability to solve tasks with deep dependency structures (C). For this, we modified the Treasure World setup slightly: it now contains four copies of the four[4] different object types, namely `key`, `lock`, `door` and `chest`, which need to be collected in this given order (see Figure 1). The vector of pseudo-rewards corresponding to these tasks is illustrated in Figure 3, which also shows how these are conditioned on precise sequences in the inventory: to trigger a reward for the `door` task, the previous two entries must be `key` and `lock`, in that order (e.g., second row of the inventory stack). Any object picked up out of order breaks the chain, and requires starting with the `key` again.

This setup captures some essential properties of difficult continual learning domains: multiple tasks are present in the same domain simultaneously, but they are not all of the same difficulty – instead they form a natural ordering where improving the competence on one task results in easier exploration or better performance on the subsequent one. Even if as designers we care only about the performance on the final task (`chest`), continual learning in the enriched task space is more effective than learning only on that task in isolation. The bar-plot in Figure 1 shows this stark contrast: compare the height of the violet bars (that is, performance on the `chest` task) for 'unicorn(`chest`)' and 'expert(`chest`)' – the Unicorn's continual learning approach scores $4.75$ on average, while the dedicated expert baseline scores $0.05$, not better than random.

Figure 1, and in more depth, Figure 14 in the appendix, show the temporal progression of the continual learning. Initially, the agent has a very low chance of encountering a `chest` reward (flatlining for 200M steps). However, a quarter of the Unicorn's experience is generated in pursuit of the `key` goal; an easy task that is quickly learned. During those trajectories, there is often a `key` on top of the inventory stack, and the agent regularly stumbles upon a `lock` (about 4 times per episode, see the gray curve on the second panel of Figure 14). While such events are irrelevant for the `key` task, the off-policy learning allows the same experience to be used in training $Q(s, a; g_{\text{lock}})$ for the `lock` task, quickly increasing its performance. This same dynamic is what in turn helps with getting off the ground on the `door` task, and later the `chest` task. Across 5 independent runs, the `chest` expert baseline was never better than random. On the other hand, the glutton baseline learned a lot about the domain: at the end of training, it collects an average of $38.72$ `key`, $16.64$ `lock`, $6.05$ `door` and $1.05$ `chest` rewards. As it is rewarded for all 4 equally, it also encounters the same kind of natural curriculum, but with different incentives: it is unable to prioritize `chest` rewards. E.g., after a sequence (`key`, `lock`, `door`), it is equally likely to go for a `key` or a `chest` next, picking the nearest. This happens at every level and explains why task performance goes down by a factor 2 each time the dependency depth increases (Figure 1). In contrast, the Unicorn conditioned on $g_{\text{chest}}$ collects an average of $9.93$ `key`, $6.99$ `lock`, $5.92$ `door` and $4.75$ `chest` rewards at the end of training. A video of the Unicorn performance in the 16 object setup and when solving Treasure World can be found in the supplementary material and is available online[5].

**Ablative study - scalability:** Additional experiments on a simpler version of the experiment with only 3 objects can be found in the appendix (Figures 12 and 15). In that case, the single-task expert baseline eventually learns the final task, but very slowly compared to the Unicorn. The difficulty increases steeply when going from 4 to 5 strictly dependent tasks (adding `cake` after `chest`), see Figures 13 and 16 in the appendix.

**Preliminary results - Meta-learning**. Closely related to the choice of how to act in the presence of many goals is the choice of what (pseudo-rewards) to learn about, which can be seen as a form of learning to learn, or meta-learning (Schaul & Schmidhuber, 2010). In particular, an automatic curriculum (Karpathy & Van De Panne, 2012; Graves et al., 2017; Heess et al., 2017; Sukhbaatar et al., 2017; Riedmiller et al., 2018) with progressively more emphasis on harder tasks can accelerate learning via more efficient exploration and better generalization (Bengio et al., 2009; Florensa et al.,

---

[4]With only 16 (respawning) objects in total, this is less dense than in the experiments above, in order to make it less likely that the agent collects an out-of-sequence object by mistake.

[5]https://youtu.be/h4lawNq2B9M

2017; Graves et al., 2017). In preliminary experiments (not shown), we attempted to do this by letting a bandit algorithm pick among goal signals in order to maximize the expected squared TD error (a proxy for how much can be learned). We found this to not be clearly better than uniform sampling, consistent with Graves et al. (2017).

## 5 DISCUSSION

**Richer task spaces**. The related work on successor features (Barreto et al., 2017) devised a particularly simple way of defining multiple tasks within the same space and decomposable task-dependent value functions. Incorporating successor features into the Unicorn would make it easy to specify *richer task interactions* where the goals may be learned, continuous, or have 'don't care' dimensions.

**Discovery**. Our setup provides explicit task definitions and goal signals to the agent, and thus does not address the so-called discovery problem of autonomously decomposing a complex tasks into useful subtasks. This challenging open area of research is beyond our current scope. One promising direction for future work would be to discover domain-relevant goal spaces using *unsupervised* learning, as in the 'feature control' setting of UNREAL (Jaderberg et al., 2016) or the $\beta$-VAE (Higgins et al., 2016).

**Hierarchy**. Our method of acting according to uniformly sampled goal signals is too simple for richer goal spaces. Fortunately, this problem of intelligently choosing goal signals can be framed as a higher level decision problem in a *hierarchical* setup; and the literature on hierarchical RL contains numerous approaches that would be compatible with our setup (Sutton et al., 1998; Bacon & Precup, 2017; Mankowitz et al., 2016; Vezhnevets et al., 2017; Kulkarni et al., 2016); possibly allowing the higher-level agent to switch goal signals more often than once per episode.

**Agent improvements**. Additions include optimizing for risk-aware, (Tamar et al., 2015), robust (Tamar et al., 2014; Mankowitz et al., 2018) as well as exploratory (Bellemare et al., 2017; O'Donoghue et al., 2017) objectives to yield improved performance. Combining *Rainbow*-style (Hessel et al., 2017) additions to the UVFA objective as well as auxiliary tasks (Jaderberg et al., 2016) may also improve the performance.

## 6 CONCLUSION

We have presented the Unicorn, a novel agent architecture that exhibits the core properties required for continual learning: task generalization and off-policy learning. Unicorn is able to (A) efficiently learn about multiple tasks, (B) leverage learned knowledge to solve related tasks, even with zero-shot transfer, and (C) solve tasks with deep dependencies. All of this is made efficient by off-policy learning about multiple tasks simultaneously, using parallel streams of experience coming from a distributed setup. Experiments in a rich 3D environment indicate that the Unicorn clearly outperforms the corresponding single-task baseline agents, scales well, and manages to exploit the natural curriculum present in the set of tasks.

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

# A   UNICORN ALGORITHM

---

**Algorithm 1** Unicorn algorithm

---

**Require:** $\theta$ - Unicorn network weights, $K$ - number of goals, $M$ - number of actors, $l$ - episode length, $H$ - unroll length which is the number of steps for n-step update ($l \gg H$), $T$ - number of episodes for training, $L$ - how often to perform learning, $B$ - batch size for learning, $Queue$ - a global queue for storing trajectories of length $H$.

1: $Queue \leftarrow \emptyset$                           ▷ Initialize the queue
2: Initialize $Q_\theta(s,a,g)$
3:
4: **while** $episode < T$ **do**
5:      **for** $m = 1 : M$ **do**           ▷ Episode start: Sample a new goal for each actor
6:          $g_m \sim Uniform(1, \cdots, K)$             ▷ Sample from $K$ goals
7:      **end for**
8:
9:      **Acting: For each actor $m$ in parallel:**
10:      Generate a $\langle s, a, r, s' \rangle$ trajectory of length $l$ using $Q_\theta(s,a,g_m)$
11:      Every $H$ steps, where ($H << l$), append the $H$ length sub-trajectory to $Queue$
12:      $episode+ = 1$                     ▷ Increment the episode count
13:
14:      **Learning:**
15:      **if** $episode\%L == 0$ **then**          ▷ Every $L$ steps, perform learning update
16:          Sample a batch of $B$ trajectories from the queue
17:          (Each batch trajectory is generated from an actor conditioned on one of the $K$ goals)
18:          **for** $b = 1 : B$ **do**          ▷ Loop through each trajectory
19:              Compute the n-step return for each of the $K$ goals (including the on-policy goal)
20:              **For all off-policy goals, perform off-policy corrections:**
21:              Truncate n-step trace at all times t where the action chosen by on-policy goal
22:              $g_i$ does not match the action that the off-policy goal $g_j(j \neq i)$ would have chosen
23:          **end for**
24:          Compute Loss $\mathcal{L}$ of Equation 1 (with additional batch dimension)
25:          Compute the gradient $\nabla_\theta \mathcal{L}$
26:          $\theta_{new} = \theta + \alpha_t \nabla_\theta \mathcal{L}$          ▷ Update network weights
27:          $\theta = \theta_{new}$
28:      **end if**
29: **end while**
30:

---

# B   UNICORN ARCHITECTURE AND HYPERPARAMETERS

## B.1   NETWORK ARCHITECTURE

The neural network architecture consist of two convolutional layers (with 16-32 kernels of size 8-4 and stride 4-2 respectively). The convolutional layers are followed by a fully connected layer with 256 outputs and an LSTM with a state size of 256. A one-hot encoding of the previous action and the previous clipped reward are also concatenated to the input to the LSTM.

The output of the LSTM layer is concatenated with an inventory $I \in \mathbb{R}^{5L}$, which is a single vector, augmented with the one-hot representations of the last five objects collected by the agent. The parameter $L$ is the length of the dependency chain. This yields the feature vector $f(s) \in \mathbb{R}^{256+5L}$ as described in the main paper. At this point, we input the entire set of goals into the network as a $K \times d$ one-hot matrix where $K$ is the number of goals, and $d$ is the dimensionality of the goal representation. The $i^{th}$ row of this matrix defines the $i^{th}$ goal $g_i \in \mathbb{R}^d$. We then duplicate $f(s)$ to generate a $K \times (256 + 5L)$ matrix and concatenate $f(s)$ with the goal matrix to form a $K \times ((256 + 5L) + d)$ feature matrix. This matrix is then fed through an MLP consisting of two hidden layers with ReLU activations. The hidden layer output sizes are 256 and $A$ respectively, where $A$ is the number of actions available to the agent. This yields a $K \times A$ matrix of Q-values,

where row $i$ corresponds to $Q(s, a_1; g_i) \cdots Q(s, a_A; g_i)$. At the start of each episode, each UVFA actor is randomly assigned a goal $g_j$ and chooses actions in an epsilon greedy fashion.

### B.2 Loss

We optimize the loss with the TensorFlow implementation of the RMSProp optimizer, using a fixed learning rate of $2e - 4$, a decay of $.99$, no momentum, and an epsilon equal to $.01$. The n-step $Q$ targets are considered fixed and gradients are not back-propagated through them.

### B.3 Distributed setup

200 actors execute in parallel on multiple machines, each generating sequences of 20 environment frames, batched in groups of 32 when processed by the learner.

### B.4 State and Action Space

The state observation is a rendered image frame with a resolution of $84 \times 84$, and pixel values are rescaled so as to fall between $0$ and $1$.

The actors use an 8 dimensional discrete action space:

1. Forward
2. Backward
3. Strafe left
4. Strafe right
5. Look left
6. Look right
7. Look left and forward
8. Look right and forward.

Each action selected by the actor is executed for 8 consecutive frames (action repeats is 8 - Table 1) in the environment before querying the actor for the next action. The actors select actions using an $\epsilon$-greedy policy where $\epsilon$ is annealed from 1 to $0.01$ over the first million environment frames.

### B.5 Hyperparameter Tuning

A grid search was performed across a selected set of hyperparameters in order to further improve the performance. Table 1 lists all of the hyperparameters, their corresponding values and whether they were tuned or not.

## C Videos

A video is included with the supplementary material showcasing the performance of the trained Unicorn in a multi-task and continual learning setting.

### C.1 Learning about multiple tasks

In the multi-task setting, the Unicorn conditions one of its UVFA actors on the goal of collecting *red balloons*. As can be seen in the video, the Unicorn UVFA agent searches for, and collects, red balloons. Due to the clutter in the environment, it may occasionally collect an incorrect object. The performance of the Unicorn compared to the baselines for collecting red balloons can be seen in Figure 7.

### C.2 Learning deep dependency structures

In the continual learning setting, we showcase the performance of the trained Unicorn agent in a setting with a dependency chain of length 3 (i.e. key $\rightarrow$ lock $\rightarrow$ door) and a chain of length 4 (i.e. key $\rightarrow$ lock $\rightarrow$ door $\rightarrow$ chest). In both cases, we condition the Unicorn agent on the goal corresponding to the deepest dependency (i.e. collect a door reward for the dependency chain of

Table 1: Hyperparameter settings for the Unicorn architecture.

| Hyperparameter | Value | Tuned |
|---|---|---|
| learning rate | $2e-4$ | ✓ |
| discount $\gamma$ | 0.95 | ✓ |
| batch size | 32 | ✓ |
| unroll length | 20 | ✗ |
| action repeats | 8 | ✓ |
| $\epsilon$ end value | 0.01 | ✓ |
| decay rate | 0.99 | ✗ |
| num. actors $M$ | 200 | ✗ |

length 3 and collect a `chest` reward for the chain of length 4). The Unicorn agent only receives a reward for the task if it achieved all of the prerequisite tasks in the order given above, yielding a sparse reward problem. As can be seen in the video, the agent is, in both settings, able to achieve the deepest dependency task. The performance of the Unicorn in each task can be seen in Figures 12 and 14 for dependency chains of length 3 and 4 respectively.

## D    ADDITIONAL RESULTS

We have additional figures corresponding to:

1. Learning about multiple tasks.
2. Transfer to related tasks.
3. Learning about deep dependency structures.

### D.1    LEARNING ABOUT MULTIPLE TASKS

Figure 8 presents the performance of the Unicorn compared to the various baselines discussed in the main paper for multi-task learning with 7 objects. Figure 7 presents multi-task learning curves on a subset of 4 out of the 16 objects in the multi-task learning with 16 objects setup.

### D.2    TRANSFER TO RELATED TASKS

Figure 7 shows the pure off-policy learning results where objects of a single color (cyan) were placed into a hold-out set and were only learned about from pure off-policy experience. We can see in Figure 7 (right) the ability of the Unicorn to learn from pure off-policy experience and solve the task of collecting cyan guitars; albeit in a slightly sub-optimal manner.

Figures 10 and 11 present the performance of the Unicorn agent when performing zero shot learning on four hold-out tasks for $K' = 12$ and $K' = 20$ training tasks respectively. In these cases, the Unicorn has seen no training data from any of these tasks; in contrast to the off-policy experiments detailed in the previous paragraph.

### D.3    LEARNING ABOUT DEEP DEPENDENCIES

Figure 14 presents the performance of the Unicorn compared to the baselines as a function of learning samples for a dependency chain of length 4. As can be seen in the graphs, the expert fails to get off the ground whereas the Unicorn learns to collect a significant amount of `chest` rewards.

Figure 12 presents early (500 million frames) and late (1500 million frames) Unicorn performance for the 3 chain dependency task (`key`, `lock`, `door`). The performance for each task as a function of learning samples can be seen in Figure 15. As can be seen in the graphs, the expert does reach comparable performance to that of the Unicorn, but takes a significantly longer time to converge. Figure 13 shows the learning performance of the 5 chain dependency task (`key`, `lock`, `door`, `chest`, `cake`). The performance as a function of learning samples can be found in Figure 16. As seen in the figures, the Unicorn fails to solve the `cake` task in the 5 chain dependency setup.

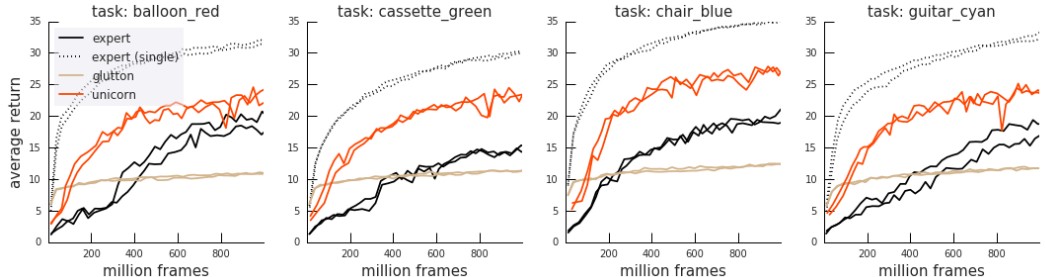

Figure 7: Multi-task learning curves, performance on a subset of 4 out of the 16 tasks, as indicated by the subplot titles. There are two lines for each color, which are from two separate seeds.

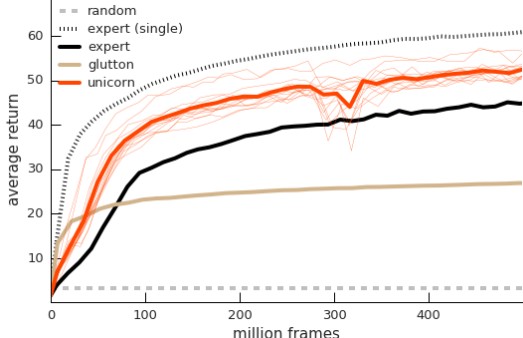

Figure 8: Multi-task learning with 7 objects. Performance is qualitatively very similar to Figure 4 with 16 objects, except that learning is overall faster.

We mention some future directions in the Discussion section of the main paper that may scale the Unicorn to solve even deeper dependency structures.

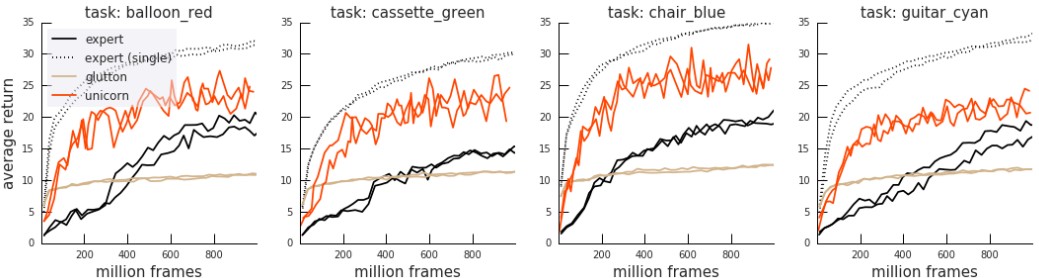

Figure 9: Off-policy transfer (I) learning curves. Only the last subplot `guitar cyan` is a transfer task. See caption of Figure 7.

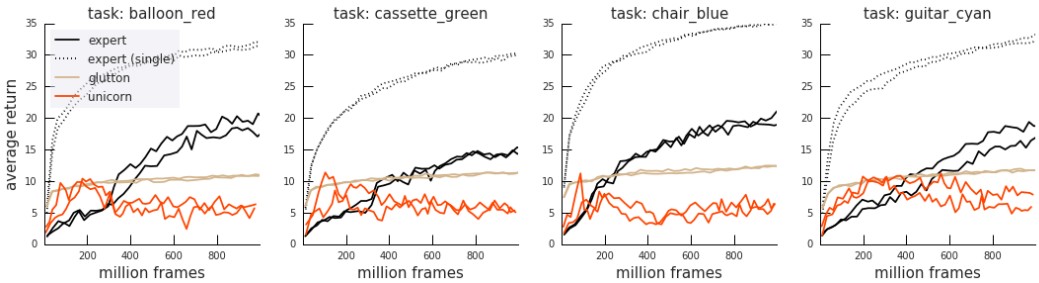

Figure 10: Zero-shot transfer (II) learning curves on all 4 transfer tasks. See caption of Figure 7.

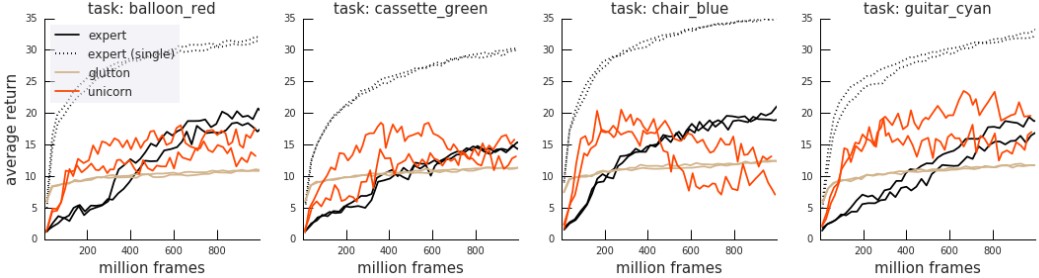

Figure 11: Augmented zero-shot transfer (III) learning curves on all 4 transfer tasks. See caption of Figure 7.

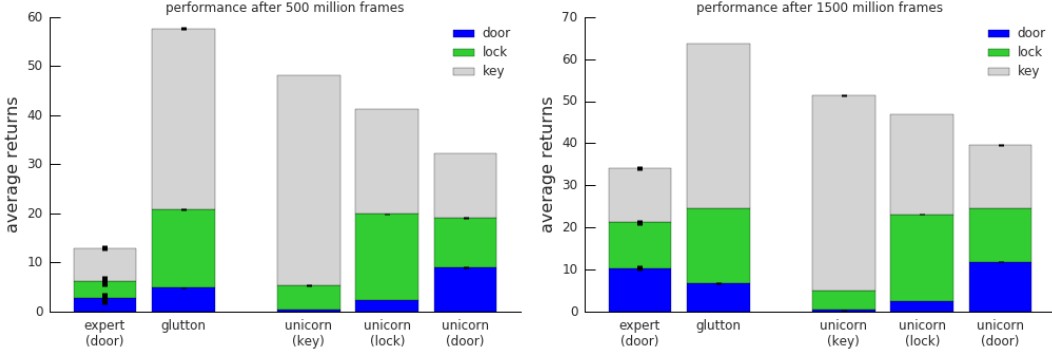

Figure 12: Continual learning with depth 3 task dependency. Early learning (left) and late learning (right). The performance gap compared to the expert is much larger in the beginning. See caption of Figure 1.

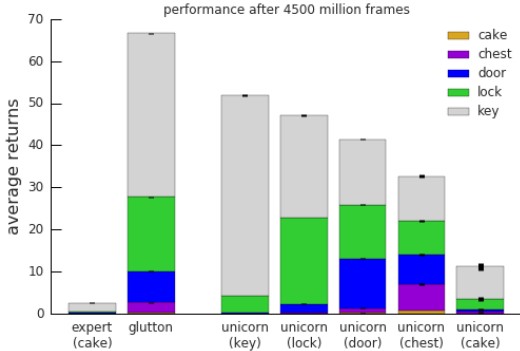

Figure 13: Continual learning with depth 5 task dependency. This is a partially negative result because the Unicorn learns to eat a lot of `cake`, but only when following the `chest` policy. See caption of Figure 1.

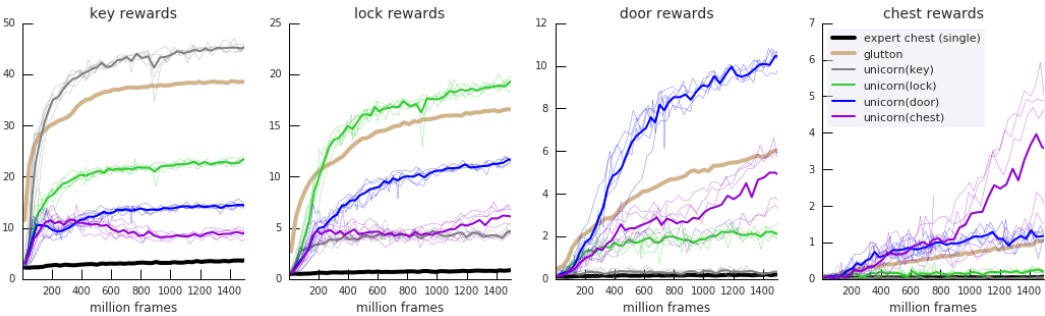

Figure 14: Continual learning with depth 4 task dependency. Performance curves as a function of learning samples. Each subplot shows the performance on one of the tasks (e.g. only counting chest rewards on the right-hand side). Thin lines are individual runs (5 seeds), thick line show the mean across seeds. In all 4 subplots the expert (black) is a single-minded baseline agent maximizing the (very sparse) chest reward only.

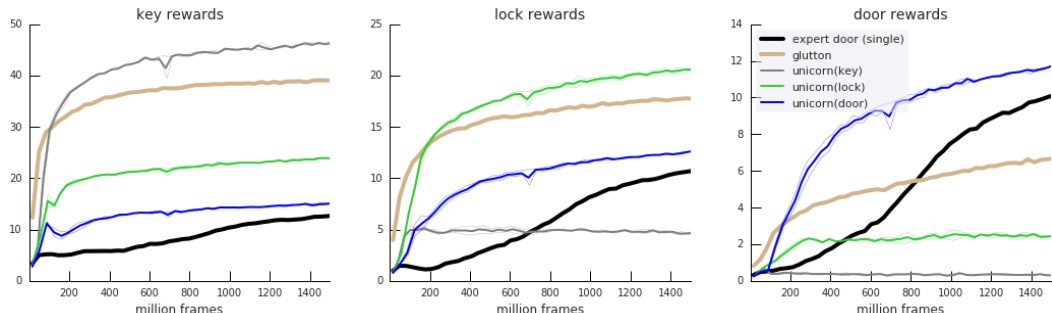

Figure 15: Continual learning with depth 3 task dependency. Performance curves as a function of learning samples. See caption of Figure 14.

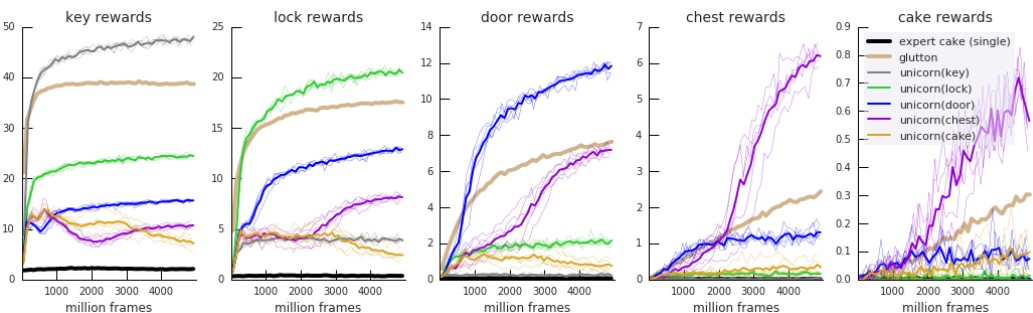

Figure 16: Continual learning with depth 5 task dependency. Performance curves as a function of learning samples. See caption of Figure 14.

