# OpenReview forum: "Unicorn: Continual learning with a universal, off-policy agent"
_ICLR.cc/2019/Conference_

### Official Review · AnonReviewer3 · 2018-10-27
**Interesting approach, but explanation can be clearer, and scope is limited.**

**Rating:** 6
**Confidence:** 4

**Review:**

Summary

The authors aim to do continual learning to solve dependent tasks using "single-stage end-to-end learning". The resulting "Unicorn" agent trains on all tasks simultaneously. The idea is to use multi-task "off-policy learning", which uses (old) trajectories (experience) from task A to help learning on a related task B. Authors further distinguish between goals (inputs to Q) and tasks (different reward functions). A goal might a color/shape of an object to pick up.

The core model is a UVFA that learns a goal-conditioned Q-function Q(s,a,g).

Some technical aspects:
- use n-step returns.
- when training Q on goal g_i, authors also trajectories that were generated using Q conditioned on another goal(s) g_j. They then truncate the returns for task i when an action taken is not optimal under Q(s,a,g_j) conditioned on goal j. The intuition (seems) to be that this
- authors do not use experience replay or a target Q-function, since the parallelized implementation is reported to be stable enough.
- unicorn sees all train task reward functions during training (but not hold out task rewards).
- unicorn is tested on several 3d maze environments with key-lock etc semantics. The tasks / goals seem simple, and the dependency is defined by changing colors / shapes of objects to be picked up. Authors argue unicorn has to learn to relate task rewards to these goal features.
- unicorn is compared against baselines that 1) do single-task learning (expert) 2) learn on a sum of task rewards (glutton), 3) uniformly random baseline.
- authors show that 1) unicorn performs better on train tasks 2) performs better on hold-out tasks. Also, authors show results for zero-shot transfer learning, with adding abstract tasks (extra reward for picking up any object) improving performance,

Pro
- Simple approach (e.g., no experience replay etc), and uses only a limited set of techniques (e.g., reward truncation).
- Reward performance suggests the model has more properly related goal features to different payoffs.
- Analysis of qualitative behavior is nice.

Con
- The writing is a bit dense in places, e.g., the discussion of baselines is a bit hard to read.
- Description of algorithm is wrapped in long text, a clear algorithm box would make the approach much clearer.
- Not clear what kind of hyperparameters are introduced / used / tuned for Unicorn.
- Authors say "deep dependency", but this seems to just refer to different colors / shapes between objects in the env used in the paper. How is "dependency" between goals and tasks defined in general?
- The experimental setting seems a bit limited, authors only show results on a single domain, and do not offer rigorous definitions. This makes the scope of the paper rather limited.

Reproducibility:
- It's not clear what the variance in the baseline performance is (variance only shown for unicorn).

---

> ### Author Response · Authors · 2018-11-23
> **Response: Reviewer 3**
>
> We thank the reviewer for the feedback.
>
> *Presentation:*
> “- The writing is a bit dense in places, e.g., the discussion of baselines is a bit hard to read.
> - Description of algorithm is wrapped in long text, a clear algorithm box would make the approach much clearer.”
>
> > Thank you for the feedback. We have incorporated your suggestions into the paper. Specifically, we have clarified our contributions in the introduction, made the baseline description clearer and have added a full algorithm to the Appendix.
>
> “- Not clear what kind of hyperparameters are introduced / used / tuned for Unicorn.”
>
> > Please see our Appendix for the details.
>
> *Deep Dependencies:*
> “- Authors say "deep dependency", but this seems to just refer to different colors / shapes between objects in the env used in the paper. How is "dependency" between goals and tasks defined in general?”
>
> > Dependency here refers to collecting objects in a particular order to receive a reward. This means collecting keys (TV), then unlocking a lock (ball), open door (balloon) and get the chest (cake). Only if objects are collected in this order does the agent receive a reward for collecting the cake object. If this order is violated (e.g., collects a balloon before a ball) anywhere in the dependency chain, then the agent has to start the chain again (i.e, find TVs). Please also see our *Baselines* response to reviewer 2.
>
> *Experiments:*
> “- The experimental setting seems a bit limited, authors only show results on a single domain, and do not offer rigorous definitions. This makes the scope of the paper rather limited.”
>
> > While we would ideally like to add more baselines, there are no Reinforcement Learning agents that can solve these kind of dependency chains. We therefore compared variants of state-of-the-art, parallelized value-based learning architectures to our own. The single domain is very challenging since it is both 3D and high dimensional. Numerous papers use variants of this domain for comparison, and we therefore chose it accordingly .
>
> “- It's not clear what the variance in the baseline performance is (variance only shown for unicorn).”
>
> > The variance of the baselines was omitted since it was negligible (of the order +-2 objects maximum across all experiments for the baseline agents) and would only make the graph more difficult to read.

---

### Official Review · AnonReviewer2 · 2018-11-02
**Deep dependent continual leaning scenario looks interesting.**

**Rating:** 5
**Confidence:** 4

**Review:**

The paper proposes a novel architecture in the context of multitask learning and deep dependency structure. They try to solve their target issue in reinforcement learning, thus utilize the useful architecture UVFAs (Schaul et al.) since the model is effective to manage multiple policies into single value function.

The paper is easy to follow, but with quite long explanation. They verify their architecture UNICORN in various learning scenarios, i.e., multitask-learning, generalization to related tasks, and continual learning. And I have several remarks,

- In the learning scenario, I confused that it looks a little bit far from the common definition of continual learning which is referred in machine learning area(learning various tasks in sequence, not simultaneously) but is more like an attribute learning or something.

- It is ambiguous to ensure that the proposed architecture can show state-of-the-art or comparable performance. The compared baselines, such as glutton and random, look too simple and show minor performance as already described in the paper.

The deep dependency structure setting is quite interesting, also there are many useful discussion and imposing experiments. It would be great to see much clear competitiveness and identity of the architecture.

---

> ### Author Response · Authors · 2018-11-23
> **Response: Reviewer 2**
>
> We thank the reviewer for the feedback.
>
> *Continual Learning:* While there are different definitions for continual learning (also referred to as lifelong learning), there is no requirement that an agent has to learn about each task in a sequential fashion. The agent can learn sequentially about multiple tasks in parallel and use this knowledge to solve subsequent tasks. One example is where a person learns to talk and walk at the same time. Here there are two tasks that the person can learn in parallel. The human can then leverage this knowledge to quickly learn to run while talking (the subsequent task).
>
> Another example, is if an agent is presented with a complex task that needs to be solved  (e.g, bake a cake). The task consists of multiple sub-tasks and therefore a dependency hierarchy is established (e.g., to bake a cake, we need to (1) mix flour and egg to make dough and (2) mix butter and sugar to make icing). In this example, we are learning about mixing for two different types of sub-tasks and once we have mastered this skill, we can bake a cake.
> It is our agents *unique* ability to solve multiple tasks at once, and simultaneously learn from the other tasks off-policy data, that facilities faster continual learning.
>
> “- It is ambiguous to ensure that the proposed architecture can show state-of-the-art or comparable performance. The compared baselines, such as glutton and random, look too simple and show minor performance as already described in the paper.”
>
> *Baselines:*
> For the multi-task and transfer experiments, we compared the agent to a single-task expert baseline and an expert baseline respectively. This is in addition to the glutton and random baselines. We have rewritten the baselines section to make the this section clearer.
>
> For the deep dependency (continual learning) experiments, it is important to note that this is the first *Reinforcement Learning* architecture to solve tasks with these deep dependency structures at scale in a high dimensional 3D environment. As such, there were no relevant comparable baselines other than a glutton (state-of-the-art, parallelized, n-step value-based agent which is rewarded for collecting any object) and cake expert agent (state-of-the-art, parallelized, n-step value-based agent trained to only receive reward upon collecting a cake, after having collected the previous objects in the dependency chain in the pre-defined order).

---

### Official Review · AnonReviewer1 · 2018-11-05
**Presentation hard to parse.**

**Rating:** 4
**Confidence:** 5

**Review:**

I tried to parse the paper's details multiple times but it really seems like a hard task. From what I understand, the paper is doing off-policy learning across several environments. The actors are parallelized across the environments (tasks), collecting rollouts, and the update to the learner is done on a GPU which trains on all these rollouts asynchronously collected. The learner uses a UVFA with an LSTM as its architecture. The authors learn on a set of training environments with various goals (associated with picking specific objects in an order) and test this policy's ability to work on new environments with a different set of goals.

Overall, I find the writing really a pain to parse. I wish the authors directly wrote what they are doing quickly: "We take Schaul's UVFA, make it recurrent, use the IMPALA set up of Esspholt, and show generalization to new combinations of objects to be collected as goals".

I am still trying to evaluate the paper, but for now, my rating for this is low given that the main novelty in the paper: the environments, the evaluations, the tasks are so unclear because of the verbose presentation style on trying to tell us what we already know, such as goal-conditioned learning, off-policy learning, IMPALA, etc.

---

> ### Author Response · Authors · 2018-11-23
> **Response: Reviewer 1**
>
> We thank the reviewer for the feedback.
>
> “I wish the authors directly wrote what they are doing quickly: "We take Schaul's UVFA, make it recurrent, use the IMPALA set up of Espeholt, and show generalization to new combinations of objects to be collected as goals".
>
> *Presentation:* We understand that there is a lot of information to parse in this paper. We have rewritten parts of the paper which include clarifying our contribution in the introduction, rewriting the baseline description and adding a full algorithm in the Appendix.
>
> Although we agree that the presentation is a bit dense, this is natural for a paper of this type. There are a multitude of terms that we need to describe. We combine many well-known techniques (UVFA, off-policy learning, off-policy corrections, parallel learning) to solve *continual learning problems at scale*. It is this novel combination of techniques that makes our work unique and we therefore had to explain each of these concepts in detail. The continual learning aspect is *crucial* to the novelty of this paper and we therefore had to discuss this too. It is expected that a reader familiar with the topic will skip a number of these sections.
>
> *Connection to IMPALA:* We note that the architecture we used is not the standard setup of Espehol et al.'s IMPALA. We had to make a number of modifications to the standard architecture, which illustrates the point above regarding our contributions. We had to make a number of adaptations which include (1) Converting the architecture, and algorithm to be value-based (IMPALA is an actor critic policy search architecture). (2) Adapt the architecture and loss to perform off-policy learning (UVFA) and (3) add value-based off-policy corrections. We have noted this in our contributions in the updated version of the paper.
>
> *Experiments toward Continual Learning:* We don’t just generalize to new combinations of goals. We think this is an unfair statement that detracts from the numerous experiments and ablative studies that we ran. In short, we show the following:
>
>  - Learning about multiple tasks simultaneously + ablative study to determine how the method scales
>  - Ablative study: Learning from purely off-policy data (never executing a policy to collect cyan objects)
>  - Transfer experiments to learn on new combinations of objects that it has never trained on before
>  - *The most important contribution of this paper:* Utilizing the Unicorn architecture to solve tasks with deep dependencies. This final point has never been performed at scale with an RL agent. The agent is able to collect objects with four dependencies where a reward is received only if the agent collects a key, door, chest and cake (in that order).
>  - Preliminary ablative experiment of a bandit manager that determines which goal to achieve based on the squared TD loss for each goal.

---

### Meta-Review · Area_Chair1 · 2018-12-14
**Significant concerns with current presentation clarity**

**Confidence:** 3
**Recommendation:** Reject

**Metareview:**

The authors present an interesting approach but there were multiple significant concerns with the clarity of the presentation, and some concern with the significance of the experimental results.